# Extracellular Matrix-Based and Electrospun Scaffolding Systems for Vaginal Reconstruction

**DOI:** 10.3390/bioengineering10070790

**Published:** 2023-07-01

**Authors:** Saeed Farzamfar, Elissa Elia, Megan Richer, Stéphane Chabaud, Mohammad Naji, Stéphane Bolduc

**Affiliations:** 1Centre de Recherche en Organogénèse Expérimentale/LOEX, Regenerative Medicine Division, CHU de Québec—Université Laval Research Center, Québec, QC G1J 1Z4, Canada; saeed.farzamfar@crchudequebec.ulaval.ca (S.F.); elissa.elia@crchudequebec.ulaval.ca (E.E.); megan.richer@crchudequebec.ulaval.ca (M.R.); stephane.chabaud@crchudequebec.ulaval.ca (S.C.); 2Urology and Nephrology Research Center, Shahid Beheshti University of Medical Sciences, Tehran 1666677951, Iran; naji_m_f@yahoo.com; 3Department of Surgery, Faculty of Medicine, Laval University, Québec, QC G1V 0A6, Canada

**Keywords:** vaginal tissue engineering, self-assembly, electrospinning, tissue decellularization

## Abstract

Congenital vaginal anomalies and pelvic organ prolapse affect different age groups of women and both have significant negative impacts on patients’ psychological well-being and quality of life. While surgical and non-surgical treatments are available for vaginal defects, their efficacy is limited, and they often result in long-term complications. Therefore, alternative treatment options are urgently needed. Fortunately, tissue-engineered scaffolds are promising new treatment modalities that provide an extracellular matrix (ECM)-like environment for vaginal cells to adhere, secrete ECM, and be remodeled by host cells. To this end, ECM-based scaffolds or the constructs that resemble ECM, generated by self-assembly, decellularization, or electrospinning techniques, have gained attention from both clinicians and researchers. These biomimetic scaffolds are highly similar to the native vaginal ECM and have great potential for clinical translation. This review article aims to discuss recent applications, challenges, and future perspectives of these scaffolds in vaginal reconstruction or repair strategies.

## 1. Introduction

Tissue engineering has emerged as a new treatment option with the capability of revolutionizing current modalities by restoring or enhancing tissue and organ functions. This approach combines principles from biology, biomaterials science, engineering, and medicine to develop tissue grafts that aim to regenerate damaged or diseased tissues.

Congenital vaginal anomalies (CVA) significantly affect patients’ quality of life, affect fertility, and have psychological impacts [1,2,3,4,5,6]. While surgical and non-surgical treatment options are available for treating vaginal defects, the low efficacy and long-term complications of these treatments necessitate developing alternative treatment options [7,8]. For example, repeated vaginal dilation is not convenient for the patients, and the surgical development of neovagina using tissue grafting is challenged by the unavailability of suitable donor tissues, graft failure, and the need for additional surgery for graft harvesting [9,10].

Pelvic organ prolapse is a prevalent medical condition among elderly women, characterized by the dislocation of pelvic organs or the protrusion of the vaginal wall. This condition is caused by changes in the supportive connective tissues or the surrounding muscles [11,12,13]. However, many other factors, including aging, genetics, diabetes, and obesity, influence its progress [14,15,16]. While polypropylene (PP) mesh could potentially provide structural support for vaginal wall augmentation, its non-biodegradability has been associated with challenges in its clinical application, such as erosion, chronic pain, and inflammation [2,17,18].

Fortunately, tissue-engineered scaffolds have brought new possibilities for the next generation of treatment strategies. The ultimate goal of this technology is to create an environment that resembles the extracellular matrix (ECM) for vaginal cells to adhere, secrete ECM, promote angiogenesis, and undergo remodeling by host cells [10,19]. In this context, various scaffold fabrication methods have been developed to closely mimic the native vaginal tissue’s ECM [20,21,22,23].

Currently, significant strides have been made toward developing scaffolds that mimic vaginal ECM’s fibrous structure or its composition [24,25]. Different scaffold fabrication methods have been used, including electrospinning, self-assembly, and tissue decellularization [26,27]. In the context of vaginal tissue engineering, the scaffolds produced by these methods provide many of the cues that are already expressed in the vaginal tissue matrix [28,29]. When it comes to electrospun scaffolds, their remarkable surface-to-volume ratio presents great potential for surface customization through the incorporation of diverse physical and chemical signals. These modifications serve as effective means to regulate cellular behavior [30,31]. This review discusses the applications, challenges, and future prospects of ECM-based scaffolds, including self-assembled tissues and decellularized scaffolds, as well as the electrospun constructs that mimic the architecture of vaginal ECM in the field of vaginal tissue engineering.

## 2. Vaginal Reconstruction Strategies

Different strategies have been applied to restore the lost function of the vagina by augmentation, neo-vagina creation, and the implantation of supporting structures for the improvement of vaginal wall integrity. Dilators were utilized in the development of nonsurgical techniques to enlarge the vaginal canal in cases of agenesis [32,33]. The benefits of these methods are a high satisfaction rate and preservation of native vaginal mucosa in the enlarged vagina, which is integral for the normal function of this tissue. Nevertheless, these techniques are traumatizing, time-consuming, and limited to sexually mature patients [34]. Surgical treatments are necessary for extensive reconstructions of congenital malformations and acquired dysfunctions. Surgical vaginoplasty is based on heterotopic autograft from various anatomical sites such as skin, oral mucosa, peritoneum, bowel segments, and vulvar flaps. Contrary to nonsurgical techniques, these methods are preferred for pediatric patients. However, donor site morbidity and a lack of sufficient grafts impede their widespread clinical use. Furthermore, heterologous tissue can cause several complications, including dryness and hair growth for skin grafts, and excessive mucus production for bowel segments [23,34,35].

Autologous tissues and mesh implants are frequently used for the surgical repair of pelvic organ prolapse; however, the high recurrence rate of treated cases with autologous tissues has made mesh implants the superior choice for this aim [36,37]. Reconstructive surgeries using mesh implants try to support injured pelvic tissues and improve long-standing surgical outcomes. In this regard, non-degradable PP mesh were the most widely utilized material by clinicians. However, frequently reported long-term complications of PP mesh, including pain, erosion/exposure, and infection finally ended in the American Food and Drug Administration’s withdrawal of this product [38]. On the other hand, synthetic and biologic biodegradable meshes can offer good biocompatibility, but their progressively reducing mechanical properties do not guarantee sufficiently strong new tissue formation. Consequently, novel and innovative strategies should be devised for more efficient treatment of pelvic organ prolapse [10,39].

Tissue engineering has attracted extensive attention during the last two decades as a promising strategy for the treatment of an array of human diseases that current techniques cannot offer an efficient remedy. Contrary to surgical methods, reconstruction of the vagina by tissue engineering has the merit of being tissue specific. Therefore, the native and integral features of the vagina can be achieved [40,41]. Utilizing cell-laden scaffolds for tissue engineering of the vagina is a generally accepted notion in this context. Autologous vaginal cells (vaginal epithelial and smooth muscle cells) and mesenchymal stromal cells are the most frequently used cells [41,42].

Naturally derived scaffolds owing to their favorable biocompatibility were extensively used for vaginal reconstruction. However, their weak mechanical properties, low reproducibility in production, and difficult processability have limited their clinical applications in vaginal defect repair [43]. So, innovative fabrication methods have been employed to construct scaffolds from natural and synthetic biomaterials for vaginal tissue engineering [44,45]. The current status of vaginal tissue engineering demands new fabrication methods and biomaterials to mimic the vaginal ECM. In addition, large preclinical and clinical studies are required for a detailed evaluation of each construct.

## 3. Brief Histology of Vagina and Biology of ECM

The vaginal wall does not have glands but consists of a mucosal layer and adventitia. The vaginal epithelium is a stratified squamous layer that is stimulated by estrogen to produce glycogen. When these cells desquamate, the released glycogen is converted into lactic acid, resulting in a relatively low pH environment in the vagina [2,7,46]. This environment is impermissible for pathogenic bacteria growth and is crucial for vaginal health. The mucosal layer contains an abundant number of inflammatory cells and is lubricated via the mucosa produced by the cervical glands [46,47,48,49,50,51].

Vaginal muscle is composed of two circular and longitudinal layers in which the former is next to the mucosal layer and the latter is close to the adventitia. The adventitia layer is rich in elastic fibers, providing excellent elasticity to the vaginal canal. In addition, the adventitia layer contains veins, arteries, and nerves [52,53,54].

Native ECM comprises a well-organized 3D network of biomolecules including collagen, glycoproteins (GPs), proteoglycans (PGs), elastin, laminin, fibronectin, and other proteins that regulate cellular behavior and tissue organization [55,56,57].

PGs are composed of a core protein component decorated with glycosaminoglycans such as heparin, heparan sulfate, and chondroitin sulfate. These biomolecules provide structural support for tissues and act as a reservoir for various growth factors [58,59]. Generally, PGs take part in tissue homeostasis, growth factors sequestration, and the regulation of various signaling systems [60,61,62,63,64]. In addition, they bind to cell surface receptors and regulate various cellular functions [65,66,67,68,69].

Hyaluronic acid (HA) is a glycosaminoglycan that is not linked to any protein and is not sulfated. It maintains tissue hydration and provides structural support. In addition, HA takes part in tissue homeostasis and regeneration [70,71,72].

Collagens are the main components of the ECM. In particular, collagen type I, II, and III are the main components that support tissue’s integrity and function [12,73,74]. Collagen has a sophisticated structural hierarchy where a triple helix structure is the most defining feature of this protein. In the primary structure of collagen, glycine is present along with Proline and Hydroxyproline that together form the most common sequence of collagen [75,76,77]. The α-chains contain various numbers of tripeptides that have a triple-helix in the middle and two non-helical sections at the two ends [78,79,80,81].

Native ECM is abundant with elastin fibers that ensure mechanical support and proper functionality [82,83,84,85]. Elastases degrade elastin into various elastin-derived peptides and trigger signaling pathways that finally lead to physiological maintenance of tissues [86,87,88,89].

Fibronectin modulates the mechanical properties of the tissue through conformational changes in its molecules. In addition, this macromolecule takes part in the regulation of cell adhesion to ECM via interaction with integrins [90,91].

Laminin is a group of proteins that plays a fundamental role in cellular differentiation, and its distribution is tissue-specific among different basement membranes [91,92,93].

In the context of vaginal tissue engineering, the scaffolding system should be able to mimic this structure and provide a permissive environment for cellular behavior and ultimately provide functionality. However, due to the complexity and abundance of components, creating a tissue-engineered construct that can meet these criteria and mimic the properties of the vaginal ECM is a significant challenge [94,95,96]. For instance, the biomechanical cues in the native ECM play a fundamental role in driving cellular functions and maintaining tissue homeostasis. Replicating these cues in a synthetic construct requires careful control over the production process. However, standardization of these processes is a time-consuming and costly procedure. In this context, extensive research has been performed to produce synthetic scaffolds with tunable mechanical properties. Although significant progress has been made, the introduction of other physical cues such as surface topographies adds to the complexity of the whole process [2,97]. Besides mechanical properties, the spatial organization of the ECM components is equally important. The vaginal tissue demonstrates a specific architecture with distinct layers, such as the epithelial, muscular, and connective tissue layers. These structures have different cell types and densities, and compositions, which affect the cellular behavior and functionality. Reproducing this complex network of polymers and bioactive molecules in a repeatable manner is a challenging task [2,42,98]. Another crucial issue to address is the incorporation of appropriate signaling cues. The ECM provides a myriad of biochemical signals (called niche) that control cellular functions. In this regard, various signaling molecules act in an orchestrated way to maintain tissue functionality. Incorporating these signaling cues into the scaffold is vital for guiding cellular behavior and promoting tissue regeneration [41,99,100,101,102]. On the other hand, spatiotemporal control over these signaling cues’ release also affects tissue regeneration. However, developing controlled drug release systems for the sustained delivery of these cues is another challenge [103,104].

### 3.1. Self-Assembly Method

The self-assembly method is an interesting technique for developing tissue-engineered scaffolds, which relies on the natural capability of mesenchymal cells to produce and assemble ECM components. This method comprises several steps that finally lead to the formation of a functional scaffold capable of supporting tissue regeneration. The production process is initiated by isolating mesenchymal cells from small biopsies. These cells are then expanded with a culture medium supplemented with ascorbic acid, typically at a concentration of 50 µg/mL. This substance is crucial for the secretion of ECM components, as it promotes collagen synthesis and ECM deposition. Over time, the cells produce ECM components and develop ECM sheets [105,106]. These constructs are then meticulously peeled off and stacked on top of each other, creating a multi-layered structure. This stacking process is essential for improving the mechanical properties of the scaffold. Once the layers are stacked, they are further cultured, allowing the layers to fuse together. This fusion process helps to create a cohesive and integrated tissue graft. During the culturing process, the mesenchymal cells within the stacked scaffolds may interact with surrounding cells and continue to produce more ECM components, further improving the scaffold’s integrity and mechanical properties.

To develop the epithelium on the self-assembled tissues, epithelial cells are cultured on the surface of the scaffolds to reach confluence. Finally, to support the maturation of the epithelium, the cell-scaffold constructs are placed at the air–liquid interface. This method allows the cells on the scaffolds’ surface to be exposed to air while still receiving necessary nutrients from the underlying culture media. The air–liquid interface allows the development of mature and functional epithelial tissues, mimicking the natural characteristics of the native tissue. The self-assembly method provides various merits for tissue engineering. It utilizes the inherent biological capability of stromal cells to develop ECM-rich scaffolds, closely mimicking the native tissues’ biological and architectural properties. The resulting constructs have demonstrated excellent biocompatibility and biodegradability, and support excellent maturation of the epithelium, making it an excellent candidate for tissue engineering and regenerative medicine applications [107,108].

ECM self-assembly is also utilized for disease modeling by using disease-relevant cells and microenvironments. Disease-specific cells are seeded onto the scaffolds, creating a 3D microenvironment that mimics the native diseased tissue ECM. Then, disease conditions are induced by introducing disease-specific factors. Various monitoring techniques such as gene expression analysis, immunohistochemistry studies, ECM remodeling, and changes in the properties of the ECM, can be used to assess the effects of different therapies on the disease. ECM self-assembly provides a powerful approach to investigating the interplay between cells and ECM in disease development and assessing potential therapeutic interventions [2,105].

Figure 1 shows a schematic illustration of producing tissue-engineered grafts with the self-assembly method.

In the context of vaginal tissue engineering, the self-assembled technique offers significant advantages over biomaterial-based approaches. This method provides tissue-specific cues for the development of a functional epithelium, and the resulting tissue exhibits histological and molecular characteristics similar to those of native vaginal tissue [27,109]. However, it is important to note that completely replicating the precise microenvironment of vaginal tissue is currently beyond the capabilities of existing scaffold fabrication technologies. Specifically, in the self-assembly method, the engineered tissue is constructed using mesenchymal and epithelial cells. Nonetheless, further investigation is required to understand the impact of inflammatory and endothelial cells’ secretome on the biological cues provided by the self-assembled scaffolds. Furthermore, the self-assembled scaffolds do not possess sufficient mechanical strength to endure mechanical forces upon sexual intercourse [2,110].

### 3.2. Tissue Decellularization

ECM components in the vaginal wall are a dynamic meshwork composed of various proteins, polysaccharides, and signaling molecules. Decellularized vaginal tissue provides three-dimensional (3D) scaffolding systems with a fibrous architecture and tissue-specific cues [111,112]. This process eliminates the major immunogenic molecules by removing the cellular compartments. As a result, the produced constructs may not elicit inflammatory responses, and the risk of graft rejection will be reduced [2,42]. Although the ECM components are highly preserved in nature and do not elicit immunological reactions, the remaining DNA molecules and Gal epitopes may cause some adverse reactions upon implantation [113,114].

In most tissues, cells are embedded within a dense network of proteins and polysaccharides, making it challenging to remove the cellular components and DNA molecules. As a matter of fact, many commercially available decellularized tissues still contain trace amounts of DNA in their structure. However, these small amounts of DNA do not compromise the therapeutic potential of these products [115,116,117].

Several methods have been developed for removing the cellular components from tissues. These methods may vary in terms of the used decellularization reagents and the way these chemicals are delivered into the tissues. Generally, chemical, physical, and biological methods have been used for this purpose (Table 1) [114,116]. As the characteristics of the tissues such as their cell density, thickness, and composition are different, the appropriate selection of the decellularization method and its optimization are of prime importance [115,116,118].

Sterilization of decellularized tissues is a crucial step for in vitro or in vivo evaluations. The appropriate selection of the sterilization method can be made based on the physicochemical characteristics of the tissues and the target application [116,136]. In this regard, gamma irradiation has been widely explored for the sterilization of various decellularized tissues. This method destroys the macromolecules of microorganisms and has a high penetration potential through relatively thick constructs [115,116,137]. On the other hand, UV irradiation is used for the sterilization of thin tissues with a larger surface. Ethylene oxide sterilizes the decellularized tissues by destroying microorganisms’ DNA and proteins. Owing to its high penetration capacity, this gas can be used for the sterilization of various tissues without causing any toxicity [138]. Alcohols are other widely used disinfectants that destroy microorganisms’ proteins. The main advantage of alcohols in tissue sterilization is that it does not affect the tissue’s ultrastructure [113]. CO_2_ laser has also been proposed for the sterilization of decellularized tissues. However, data available in the literature are limited, and its prospective applications in this area need to be explored further [113].

The effects of different decellularization methods on the ECM composition or its properties have been studied before. In this regard, Kusoglu et al. compared the effects of four different decellularization methods including freeze–thawing, peracetic acid, sodium dodecyl sulfate, and triton on the properties of lung tissue ECM. They showed that the viscoelasticity and stiffness of the hydrogels prepared by decellularized lung tissues were significantly different among groups, indicating that different decellularization methods may have varied effects on the intermolecular interactions between ECM polymers [139]. In addition, Sevastianov et al. showed that mechanical, biochemical, and biological properties of acellular cartilage tissues prepared by three different decellularization methods including freeze–thawing, supercritical carbon dioxide fluid, and ultrasound were significantly different. In all groups, glycosaminoglycan and collagen contents were reduced. The tissues prepared using an ultrasound method provided better cell viability with human adipose-derived stem cells [140]. Therefore, the selection of the decellularization method may potentially affect the potential healing outcome. Detailed discussion regarding the use of decellularized scaffolds in tissue engineering can be found in these references [122,141].

### 3.3. Electrospinning

Electrospinning technology is based on the process of spinning a charged polymeric solution under an electrical field. It has emerged as a promising method for fabricating scaffolds, enabling the production of constructs with a fibrous architecture that closely resemble the ECM of native tissues. In addition to its versatility, electrospinning allows for the tunability of scaffold properties and proves to be a cost-effective approach [142,143,144].

For electrospinning of a polymer, its solution is prepared in an appropriate solvent system and then loaded into a syringe connected to a metal needle. A feeder pump pushes the polymer out of the nozzle and a positive high voltage is applied to the needle [145,146,147]. At a certain voltage, the electrical forces in the needle form a polymeric jet by overcoming the surface tension forces. Then, the polymeric jet undergoes bending instability, leading to the whipping of the jet into multiple jets that are finally deposited onto the collector [147,148,149].

The main advantage of the electrospinning method is that the characteristics of the produced scaffolds can be easily changed by altering the fabrication parameters or the collector’s properties. In this context, various modifications have been made to the conventional electrospinning method [31,150]. Figure 2A shows the components of the conventional electrospinning machine. Despite being simple, this method is not suitable for a high-throughput scaffold fabrication process. Increasing the number of fiber jets may potentially address this issue. In this regard, previous studies have increased the number of fiber jets by using a multi-spinneret needle (Figure 2B) [142].

One of the disadvantages of electrospinning in scaffold fabrication is that most of the electrospinnable solvents may potentially compromise the biological functions of the bioactive molecules. On the other hand, biocompatible solvents cannot solubilize most synthetic biomaterials [151,152]. Fortunately, core–shell electrospinning method has been extensively explored for the production of hybrid scaffolds that can preserve the biological function of therapeutic agents. In this method, the bioactive molecules are dispersed in a polymer that is soluble in a biocompatible solvent and the second polymer is dissolved in another solvent. Finally, both polymeric solutions are electrospun through a core–shell needle [153].

**Figure 2 bioengineering-10-00790-f002:**
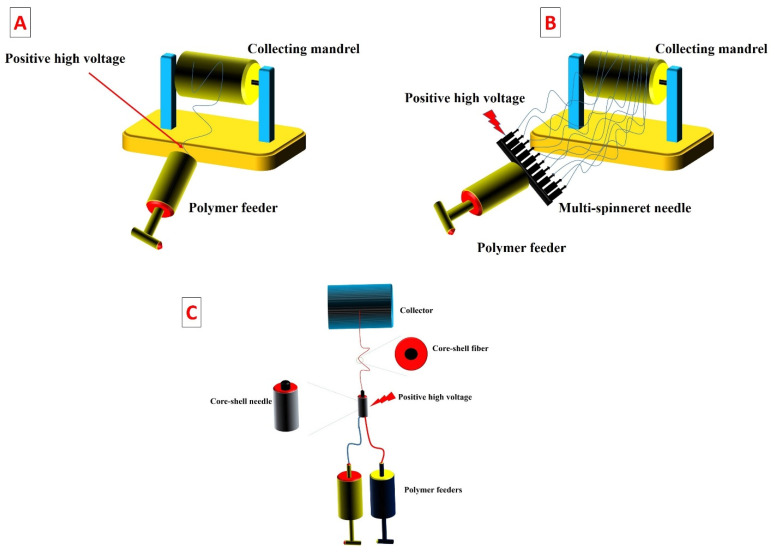
Schematic illustration demonstrating (**A**) conventional electrospinning machine, (**B**) multi-spinneret electrospinning, and (**C**) core–shell electrospinning. High positive voltage aids electrospinning by overcoming surface tension, creating an electric field that stretches polymer into ultrafine fibers. Fiber thickness varies (nanometers to micrometers) based on parameters such as concentration and viscosity. In the electrospinning method, feeders are compressed using syringe pumps to control the flow rate. The number of spinnerets affects the production yield (**B**). In addition, the core–shell electrospinning method utilizes a modified spinneret to produce chore-sheath fibrous scaffolds. Electrospinning can be performed in sterile or non-sterile conditions. Sterile conditions are essential for biomedical applications. Sterile laminar flow hoods, equipment, and sterile solutions may be used [31,150,154].

Various properties of the electrospun scaffolds can be changed by altering the production parameters including needle gauge, polymer concentration, solvent type, humidity, temperature, collector turning rate/its morphology, and polymer characteristics [155,156,157]. Table 2 summaries the effects of different parameters on the produced fiber properties.

Electrospinning can produce 3D constructs or sheets by depositing fiber jets on a collector. Sequential stacking of fibers on top of each other enables the production of scaffolds with controlled thickness. Another technique to produce 3D constructs involves electrospinning onto sacrificial supporting materials. This supporting material serves as a temporary surface for fiber deposition. Once the electrospinning is performed, the sacrificial support can be dissolved or removed, leaving behind a 3D nanofibrous scaffold. Finally, by using a collecting mandrel with specific morphology, 3D scaffolds can be produced [171,172,173].

The process of producing hybrid scaffolds in electrospinning involves the blending of different polymer types for the production of electrospun scaffolds. This approach allows for the introduction of complementary characteristics, such as improved mechanical properties, bioactivity, and drug release potential, within a single scaffold. This provides the versatility to develop advanced tissue-engineered constructs with customized properties, making them well-suited for a variety of applications [174,175,176].

Table 3 summarizes various properties of electrospinning, self-assembly, and decellularization methods in vaginal tissue reconstruction.

## 4. Preclinical and Clinical Studies on ECM-Based and Electrospun Fibrous Matrices in Vaginal Reconstruction

### 4.1. Self-Assembled Fibrous Scaffolds for Vagina Reconstruction

The current scaffolding systems for vaginal reconstruction rely on exogenous synthetic/natural polymers. These constructs lack tissue-specific biological cues, and in most cases, the vaginal epithelium is not fully differentiated. Therefore, developing a fully autologous vaginal 3D scaffold provides an excellent opportunity for vaginal reconstruction or modeling of its diseases [8,181]. In this regard, Orabi et al. developed a tissue-engineered vagina using the self-assembly method and autologous vaginal cells [181]. Vaginal stroma was prepared by culturing vaginal stromal cells in the presence of ascorbic acid. Then, three layers of stromal sheets were stacked on top of each other and allowed to fuse together. Finally, vaginal epithelial cells were seeded onto the stroma and allowed to mature at the air–liquid interface. Then, these scaffolds were implanted subcutaneously into nude mice. Histopathological evaluations showed that the developed vagina had a comparable microstructure to the native vaginal tissue. In addition, the developed vagina could integrate with the host tissues. Scanning electron microscopy images showed that the developed vagina was composed of a stratified epithelium and a cohesive stroma. Transmission electron microscopy imaging showed that the luminal surface of the vaginal equivalent had numerous microvilli and the overall structure of the developed tissues was similar to that of native vaginal tissue. This study highlights the potential applicability of self-assembled neovagina for vaginal reconstruction strategies. However, the feasibility of implementing this approach in human subjects remains in question. Specifically, the resistance of these scaffolds against mechanical forces during sexual intercourse needs to be thoroughly investigated.

One of the main challenges in tissue engineering is the limitations of diffusion to supply nutrients and oxygen in the inner sections of a 3D scaffold. In this regard, various vascularization technologies have been developed to pre-vascularize the tissue-engineered scaffolds before implanting them in vivo [182]. For instance, Jakubowska et al. used a novel method for vascularizing the self-assembled vaginal tissue [183]. They co-cultured the stromal cells and human umbilical vein endothelial cells to improve the formation of a micro-capillary network in the scaffolds. The endothelial cells tagged with green fluorescent protein and luciferase allowed tracking of the cells both in vitro and in vivo. Scaffolds were implanted into the subcutaneous tissue of female nude mice and were followed up for 3 weeks. Although the vascularized scaffolds showed some signs of the earlier formation of capillaries in vivo, statistically, no significant difference was found between graft survival outcomes in control and experimental groups. In vivo study confirmed that the pre-formed capillary-like structures could connect to the mice vasculature network and supply blood flow. Overall, this study suggests using endothelial cells alongside stromal cells in order to produce self-assembled scaffolds. In addition, using proangiogenic growth factors and signaling molecules such as vascular endothelial growth factors, insulin-like growth factors, basic fibroblast growth factors, and erythropoietin may also improve the scaffolds vascularization [184,185].

Besides using self-assembled scaffolds in vaginal reconstruction strategies, they can also be utilized in order to develop disease models. In this regard, Saba et al. produced a vaginal tissue for modeling human immunodeficiency virus type-1 (HIV-1) infection [186]. The cells were harvested from virus-negative donors and utilized to produce vaginal tissue that closely resembled the native tissue. The self-assembled matrices exhibited mechanical properties comparable to those of the native vaginal mucosa and were capable of producing glycogen. Immune competency was achieved by seeding human monocyte-derived macrophages onto the scaffolds. The immunocompetent tissue was successfully infected with HIV-1, and the viruses replicated within the tissue. Undoubtedly, this developed model holds potential as a valuable tool for studying HIV infection, virus load, virus transmission, and the development of new therapies. However, it is important to acknowledge that the complexity of immunological reactions within the body may not be easily replicated through tissue engineering approaches [187].

### 4.2. Electrospun Vaginal Matrices for Vaginal Wall Reinforcement

Women have a considerable likelihood of experiencing pelvic organ prolapse, a medical condition with limited treatment options. There is an urgent need to develop bioactive materials to support organs in the pelvic floor without compromising their functionality. In this regard, Hympanova et al. compared the healing potential of three different materials in a sheep model of vaginal wall prolapse [188]. In their study, two types of electrospun membranes, namely ureidopyrimidinone-polycarbonate (UPy-PC) and polyurethane (PU)-based matrices, were prepared and compared to native tissue repair (NTR) and ultra-lightweight PP mesh in terms of their healing efficacy. None of the electrospun scaffolds resulted in visible complications, and they did not significantly affect the contraction of the vaginal wall. Furthermore, the electrospun matrices exhibited good integration with host tissues and demonstrated well-established vascularization. Inflammation around the electrospun matrices was minimal. Although the majority of macrophage cells surrounding the implant site exhibited the M2 phenotype, known as pro-healing cells, It is important to note that the natural healing response in vaginal tissue relies on multiple signaling molecules. Therefore, incorporating these agents into the structure of electrospun scaffolds may enhance their potential for healing. Electrospun matrices are highly effective at encapsulating drugs and can provide sustained drug release, making them an excellent option for accommodating these signaling molecules. [148].

The main challenges associated with synthetic mesh applications in pelvic organ prolapse is their erosion, inflammation, and chronic pain. Chen et al. developed a new method for treating this disease using combinations of 3D printing and electrospinning methods [1]. The meshes were prepared using polycaprolacton and poly(lactic-co-glycolic acid) polymers and were loaded with lidocaine, estradiol, metronidazole, and connective tissue growth factor. The developed hybrid scaffolds exhibited mechanical properties comparable to those of commercially available PP mesh and featured a fibrous extracellular matrix-like microstructure (Figure 3A,B). Water contact angle measurements indicated that the core–shell nanofibers and drug-loaded scaffolds had significantly higher hydrophilicity compared to pure PLGA (Figure 3C–E). Additionally, transmission electron microscopy images revealed a sheath-core structural characteristic (Figure 3F,G). Animal studies demonstrated that the mechanical properties of the developed constructs decreased over time, and histopathological examinations showed no adverse tissue reactions (Figure 3H–K).

Indeed, the drug delivery potential of electrospun fibers has attracted the attention of both clinicians and researchers. Besides anti-inflammatory and analgesic drugs, various drugs are available for treating pelvic organ prolapse [189].

The vaginal ECM is composed of a web-like network of various fibers that support its structure. When this matrix is overstretched, as occurs in pelvic organ prolapse, the induction of new matrix production is crucial [190]. Vashaghian et al. hypothesized whether mechanical stimulation may improve the regenerative activity of fibroblast cell-seeded electrospun membranes [191]. Electrospun scaffolds were produced using a PCL/PLGA blend and then seeded with fibroblast cells derived from patients with pelvic organ prolapse. Once the cell-scaffold constructs reached confluence, they were subjected to cyclic strain for 24 h and 72 h. Cells exposed to mechanical strain showed a loss of myofibroblast differentiation potential. However, they exhibited upregulated expression levels of genes related to matrix production, matrix remodeling, and inflammation, suggesting that mechanical stimulation may have the potential to enhance the healing efficacy of fibroblast cell-seeded electrospun membranes. Indeed, mechanical cues play a significant role as signaling cues in the native ECM. Furthermore, in addition to mechanical cues, other biophysical cues such as scaffold stiffness, surface topography, and even electrical stimulation, should be investigated in relation to electrospun scaffolds [192].

The high surface area of electrospun nanofibers provides an excellent opportunity for their surface modification. In this regard, Verhorstert et al. explored using surface-modified electrospun poly-4-hydroxybutyrate (P4HB) scaffolds with estradiol to treat pelvic organ prolapse [193]. To test the hypothesis of whether ECM-like electrospun scaffolds can enhance the regenerative function of the scaffolds, P4HB was also knitted and compared to its electrospun counterparts. The study demonstrated that collagen deposition, elastin secretion, and cell proliferation were significantly higher in cells cultured on the electrospun scaffolds compared to those seeded on knitted constructs. Although surface modification with drugs is a feasible approach, it is not suitable for drug loading as it may result in burst drug release. An alternative approach would be to load the drugs into the matrix of electrospun scaffolds [148,194,195,196].

Wu et al. conducted a preclinical study and clinical trial to evaluate the therapeutic potential of co-electrospun poly(l-lactide-co-caprolactone) (PLCL) and fibrinogen (Fg) for pelvic organ prolapse, comparing it to the standard control, PP mesh. The researchers observed that the electrospun membranes exhibited better vascularization and were better tolerated compared to the PP group. Furthermore, patients treated with PLCL/Fg membranes demonstrated significantly greater improvements in pelvic organ prolapse quantification scores compared to the PP group. However, it is important to note that PP also provided some relief in symptoms for the patients [177].

Endometrial stromal cells are a highly proliferative source of cells that take part in uterus wall reconstruction following each menstruation cycle. Due to their high regenerative potential, their healing efficacy has been investigated in various disease models [197]. In this regard, Paul et al. bioprinted these cells onto a melted electrospun mesh to produce a construct for vaginal wall re-enforcement [26]. In vivo studies showed that cell-laden constructs integrated well into the mice tissue and preserved the viability of endometrial stem cells. In addition, the produced scaffolds promoted the recruitment of the M2 phenotype of macrophage cells around the implant. They concluded that this approach may potentially be considered as a treatment strategy for treating pelvic organ prolapse in the clinic. However, the potential undesired differentiation of endometrial stromal cells should be investigated in long-term studies [198,199].

Electrospun scaffolds have the potential to serve as a dual-purpose platform for delivering both stem cells and growth factors. In this regard, a study was conducted to assess the feasibility of using electrospun PCL scaffolds loaded with mesenchymal stromal cells and connective tissue growth factors for treating pelvic organ prolapse in elderly rats [200]. The developed scaffolds were implanted in elderly female rats and subsequently removed after 53 weeks. The researchers observed that the long-term implantation of the scaffolds led to a decrease in collagen type III synthesis, milder inflammatory responses, and histopathological changes. This study offers new insights into the potential of coating biodegradable meshes with stem cells and growth factors to reduce complications associated with long-term mesh applications.

### 4.3. Decellularized Scaffolds for Vaginal Reconstruction

Three-dimensional bioprinting technology aims at producing personalized tissues for vaginal reconstruction. In this technology, various bioinks have been developed to produce biomimetic scaffolds [201]. In a novel approach, Hou et al. developed a bioactive ink using the acellular vagina matrix [112]. The acellular scaffolds were mixed with gelatin and sodium alginate, and their physicochemical and biological properties were investigated. Subsequently, bone marrow mesenchymal stroma cells were encapsulated within the scaffolds, which were then implanted into the subcutaneous tissue of a rat model. The study demonstrated that the printed vagina exhibited good vascularization and epithelialization, without causing any adverse tissue reactions. Furthermore, it was observed that the bone marrow mesenchymal cells within the printed vagina displayed characteristics of vaginal epithelial cells and endothelial cells, indicating that the vaginal matrix may contain biological cues that direct the fate of stem cells towards vagina-specific cell types. However, despite being a promising approach for developing biomimetic scaffolds, the long-term durability of bioprinted constructs is suboptimal for clinical applications. The use of a cross-linking method can enhance their stability, but it may compromise cell viability [202].

The organ-specificity of the acellular vaginal matrix provides an overt advantage in vaginal reconstruction strategies over other methods. However, the isolation and sterilization of vaginal tissue from large vertebrates is a challenging task. Zhang et al. developed a multistep decellularization method to remove cells from the porcine vagina [203]. Then, the scaffolds were utilized to reconstruct a rat vagina model, and the results were compared with the healing effects of small intestine submucosa (SIS). The developed matrices exhibited optimal biomechanical properties and were rich in various growth factors. In the in vivo study, it was observed that the vaginal matrices were well-tolerated and integrated into the host tissues significantly better than small intestine submucosa tissue. Although this research did not report the effects of the decellularization process on the vaginal ECM, it is worth noting that decellularization methods have the potential to alter the composition or structure of the ECM.

The differentiation of fibroblast cells into myofibroblasts is crucial for repairing vaginal injuries. Previous studies have demonstrated that the vaginal ECM in patients with pelvic organ prolapse is stiffer and has a different composition compared to that of individuals without the condition. In basic research, a series of vaginal matrices with known stiffness were employed to investigate the physical properties of the matrices on the process of fibroblast-myofibroblast differentiation [96]. It seems that ECM stiffness has a positive impact on the tissue expression of α-smooth muscle actin. In addition, the differentiation was more prominent in cells seeded on vaginal ECM derived from pelvic organ prolapse patients that had a higher collagen and elastin content. Therefore, it should be noted that in developing tissue-engineered vaginas, the biophysical characteristics of these scaffolds may ultimately determine the therapeutic outcome [204].

In addition to a vagina-specific matrix, decellularized tissues from other sources may also be utilized for developing vaginal tissue. In this regard, Shen et al. seeded decellularized bladder tissue with vaginal smooth muscle cells and used it for treating vaginal defects in a rabbit model [205]. The scaffolds were seeded with smooth muscle cells at a density of 1 × 10 cells/cm^2^ and cultured for five days. The seeded cells began adhering to the scaffolds after 4 h of cell seeding, and the matrix effectively supported cellular proliferation and differentiation. In vivo studies demonstrated that 21 days after implantation, the luminal surface of the scaffolds was covered by vaginal epithelial cells. Furthermore, twelve weeks after implantation, the overall structure of the developed vagina closely resembled that of normal vaginal tissue. Vaginography studies revealed that the vaginal canal remained open without any signs of fibrosis or graft rejection. This study suggests the potential application of non-vaginal tissue matrices as a scaffolding system for vaginal tissue engineering. However, despite these promising results, the availability of donor tissues, the risk of disease transmission, immune rejection, and the long-term stability of these grafts are potential challenges associated with this strategy for vaginal defect repair [117].

The luminal collapse of the developed vaginal tissue remains a challenge. In this regard, Wefer et al. used acellular vaginal or bladder matrices for vaginal reconstruction in a rat model [206]. Although the vaginal length between the specific and non-specific matrices did not show a significant difference, the regenerative outcomes appeared to be slightly better in the organ-specific vagina matrix. In both groups, the vaginal lumen was nearly closed after 12 weeks of implantation. Therefore, new strategies should be developed to promote epithelization of the vaginal luminal surface and prevent the loss of the lumen.

## 5. Challenges, Future Perspectives, and Concluding Remarks

Despite the favorable characteristics of the scaffold fabrication methods reviewed in this review, there are constraints and obstacles to their clinical translation. With regard to the self-assembly method, the produced scaffolds may show delamination after stacking multiple layers of stromal sheets. On the other hand, due to their poor mechanical properties, they might not endure shear and tear forces upon sexual intercourse.

Most of the electrospun scaffold applications in vaginal reconstruction have been focused on pelvic organ prolapse repair. Indeed, these constructs hold great potential for drug and cell delivery into the vaginal wall. However, challenges such as a low production yield, small pore size for cell ingrowth/vascularization, the potential toxicity of residual solvents, and low processability of naturally occurring polymers need to be addressed [142]. The adverse impact of sterilization procedures on the mechanical properties of scaffolds may also pose a challenge for the clinical translation of these constructs. Finally, the successful transition of electrospun meshes from the research and development stage to a high-throughput fabrication technology necessitates the development of machines capable of achieving higher production yields. This is an essential step for the scalability and commercial viability of this technique as a manufacturing technique. The current challenges in the production yield can be due to factors such as process efficiency, equipment design, and material characteristics. Addressing these issues will require advancements in electrospinning technology, including the optimization of the fabrication parameters, the design of specialized equipment, and the exploration of novel materials that are suitable for mass production. By addressing these considerations, electrospun scaffolds can potentially be translated into the clinic for treating vaginal defects [31,155,207].

Concerning tissue decellularization, selecting the suitable tissue for decellularization, potential alterations in physicochemical properties of tissues after decellularization, the potential adverse impacts of the sterilization procedures, and challenges of quality assessment have been explored previously [114,117]. Preservation of ECM components and structure is not optimal with the current decellularization methods. Therefore, developing cell-removal methods with minimal impact on the biocompatibility of scaffolds or their structure would be beneficial.

One essential aspect of the clinical translation of these scaffolds is conforming to the regulatory safety measures. The development and implementation of electrospun, self-assembled, and decellularized scaffolds for vaginal reconstruction require rigorous testing to ensure their safe application. These constructs should undergo thorough preclinical experiments to unravel their potential adverse effects and establish guidelines for their use.

The process of developing and translating these scaffolds into clinical practice can be time-consuming and costly. Furthermore, scaling up the high-throughput production technology to meet the demands of a clinical setting can present further challenges. The time and financial investments required can be significant obstacles to widespread clinical use.

Given the complex nature of these scaffolds, developing the process of these products requires specialized knowledge and skills. Highly trained laboratory personnel with expertise in tissue engineering and regenerative medicine are crucial for successful clinical translation. The availability of such personnel may be limited, especially in healthcare settings where resources are already stretched thin.

When introducing a new treatment modality, patient selection becomes a crucial consideration. Not all patients may be eligible candidates for treatment with these scaffolds. Therefore, careful evaluation of potential risks and benefits is of vital importance. Factors such as overall health, previous medical history, and individual anatomical considerations should be considered to test the suitability of a patient for a particular technique or scaffold.

In our opinion, the ideal scaffolding system for vaginal tissue engineering should not only withstand external mechanical forces but should also provide essential cues for the proper functioning of vaginal cells. However, none of the fabrication methods reviewed in this study can produce a scaffold that can efficiently meet all of these criteria. Fortunately, the versatility of these approaches offers an opportunity to develop multifunctional scaffolding systems. In this context, combining the electrospinning method with other techniques could enhance their properties and further increase their chance of clinical translation [2].

With the progress in ECM biology, new decellularization technologies, and advances in biomaterials science, we hope that many of the current drawbacks can be addressed. We envisage more clinical trials with these scaffolding systems in vagina reconstruction.

## Figures and Tables

**Figure 1 bioengineering-10-00790-f001:**
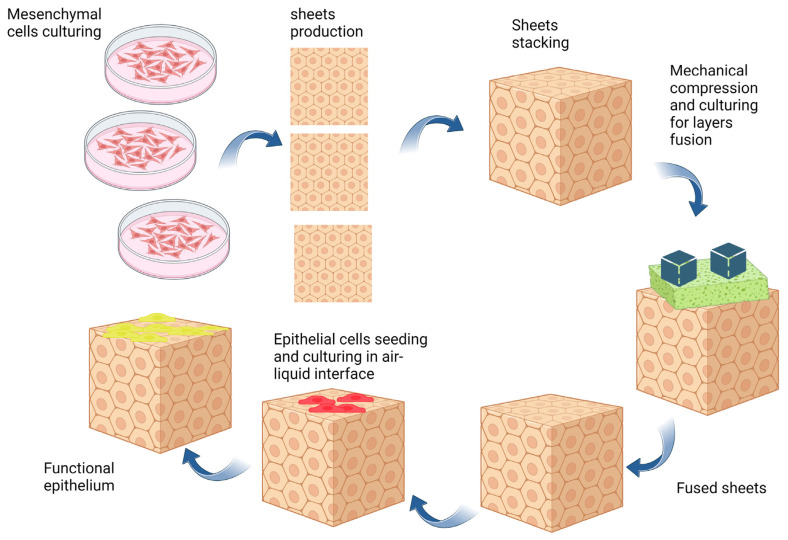
Schematic illustration representing the principles of the self-assembly method. For the production of the tissues, the mesenchymal cells are cultured in Petri dishes with a paper anchorage in the presence of ascorbate (50 µg/mL). Then, the produced ECM sheets are stacked, pinned together using surgical clips, and mechanically compressed by metal weights. The multilayer construct is further cultured to allow the fusion of the layers. Then, epithelial cells are seeded onto the constructs and cultured submerged to cover the surface. Finally, the cell-scaffold constructs are placed at the air–liquid interface that leads to the maturation of the epithelium. Adapted from Ref. [27].

**Figure 3 bioengineering-10-00790-f003:**
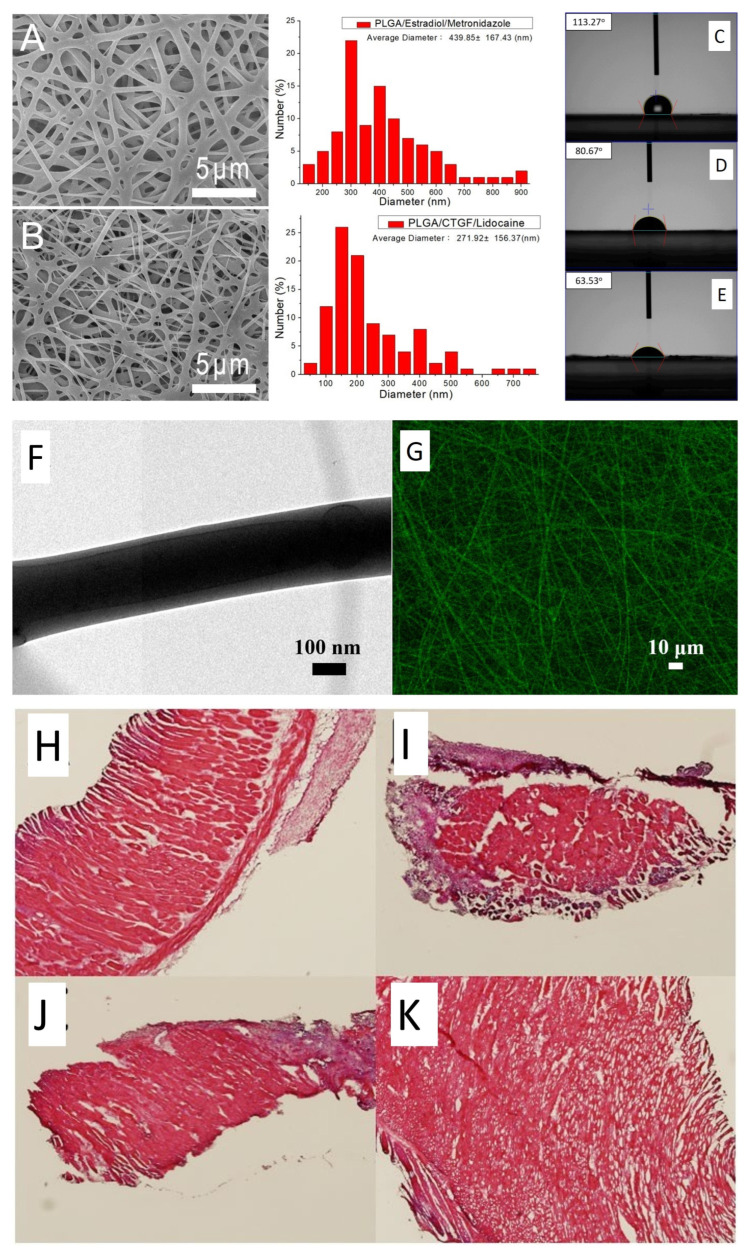
(**A**,**B**) SEM images and fiber diameter distribution of drug-loaded PLGA nanofibers and sheath-core-structured PLGA/CTGF nanofibers, respectively. (**C**–**E**) Water contact angles of PLGA, drug-loaded PLGA and sheath-core CTGF-embedded PLGA nanofibers, respectively. (**F**) TEM image and (**G**) laser scanning confocal microscope image of core–shell constructs. (**H**–**K**) Histological images at 1, 4, 7 and 28 days post-implantation, respectively. Adapted from Ref. [1].

**Table 1 bioengineering-10-00790-t001:** Summary of decellularization methods.

Decellularization Method	Category	General Characteristics	Examples	Pros/cons	References
Ionic detergents	Chemical	These chemicals solubilize DNA and cell membrane, leading to the removal of cellular components	Sodium dodecyl sulfate, sodium Deoxycholate and Triton X-200	Damaging the ECM integrity, removing growth factors and glycosaminoglycans	[119,120,121,122]
Non-ionic detergents	Chemical	These reagents weaken the interaction of lipids with other lipids or proteins. However, protein-protein interactions remain unaffected by these chemicals.	Triton X-100	The ultrastructure of ECM or its growth factor content is preserved. However, the decellularization efficacy is lower than the ionic detergents.	[122,123,124,125]
Zwitterionic Detergents	Chemical	These chemicals have similar properties with ionic and non-ionic detergents.	Sulfobetaine-10 and Tri (n-butyl) phosphate,	These chemicals have higher decellularization potential than non-ionic chemicals and preserve ECM better than the ionic detergents.	[115,119,126]
Chelators	Chemical	These agents bind to divalent metal cations and loosen the cells binding to their surrounding ECM.	Ethylene glycol tetraacetic acid (EGTA)	These reagents do not damage the ECM components.	[119,127,128]
Bases/Acids	Chemical	Extreme pH conditions damage the cells and remove them from the tissues. Acids have been found to damage cytoplasmic membrane and DNA complexes.	Ammonium hydroxide and Acetic acid	These chemicals damage the growth factors and ECM’s structure.	[120,129,130]
Alcohols	Chemical	They diffuse through the cellular membrane and damage DNA and cells via dehydration.	Methanol and Ethanol	These chemicals may affect the ultrastructure of ECM.	[120,131,132]
Hypertonic and Hypotonic Solutions	Chemical	These cells lyse the cells by disrupting the osmotic pressure.	Sodium chloride solutions.	These solutions do not remove cellular debris and are often used with other chemical reagents.	[133,134]
Enzymes	Biological	They cleave the bonds between biological macromolecules.	Phospholipase A2, proteases, and nucleases	Proteinases may damage the ECM’s structure. Nucleases are often used with other detergents to remove DNA remnants.	[120,131]
Agitation Immersion and Pressure	Physical	The physical forces caused by agitation and pressure lead to cellular damage	-	This method is often used in combination with chemical reagents to increase the exposure of cells to chemicals.	[116]
Freeze–Thaw Cycles	Physical	Freeze–thaw cycles damage the cellular membrane and cause the formation of intracellular crystals.	-	Although this method does not affect the ECM’s ultrastructure, it does not effectively remove cellular debris.	[135]

**Table 2 bioengineering-10-00790-t002:** Summary of parameters affecting the electrospun scaffolds’ properties.

Parameter	Effects	References
Polymer properties	Low-molecular-weight polymers produced beady fibers. On the other hand, high-molecular-weight polymer tends to produce uniform fibers.	[158,159]
Polymer concentration	Highly concentrated polymers produce fibers with a greater diameter.	[160]
Needle gauge	Large needles produce thicker fibers.	[161,162]
Solvents conductivity	The solvent conductivity affects the fibers’ average diameter and their morphology	[163,164]
Voltage	Higher voltages decrease fibers’ diameter and increase their crystallinity.	[165,166]
Polymer feeding rate	Higher polymer volume results in thicker fibers	[165,167]
Needle to collector distance	Short distance produces thicker fibers and vice versa.	[168,169]
Collector properties	The turning rate of the mandrel and its morphology affects the fibers’ alignment and the thickness of the produced scaffolds.	[142,170]
Environmental conditions	Humid environments may affect the solvent’s volatility and result in fiber fusion. High temperature may result in rapid evaporation of solvents and morphological change in the fibers. Air pressure affects the solvent volatility and fiber structure.	[161,170]

**Table 3 bioengineering-10-00790-t003:** Summary of various properties of electrospinning, self-assembly, and decellularization methods in vaginal tissue engineering.

Scaffold Fabrication Method	Time of Production	Area of Application	Clinical/Preclinical Studies	Similarity to Native Vagina	Weaknesses	Strengths	References
Electrospinning	Hours	Vaginal tissue reconstruction and vaginal wall reinforcement	Both clinical and pre-clinical studies have been performed.	Architectural similarities	Lack of biological cues	Ease of fabrication, suitable for mass production, excellent mechanical properties depending on the use of materials, tailorable properties	[24,31,177,178]
Self-assembly	Days	Vaginal tissue reconstruction	Pre-clinical	Composition and architecture	Poor mechanical strength	Free of exogenous materials, suitable for personalized medicine, provision of various cues for vaginal tissue reconstruction	[27,109]
Decellularization	Can range from several hours to several days, depending on the specific tissue, decellularization technique, and subsequent processing steps.	Vaginal tissue reconstruction	Pre-clinical	Composition and architecture	Poor mechanical strength and ethical issues, potential immunogenicity, risk of disease transmission	Availability of tissues from cadaveric donors and preservation of native vagina’s ECM	[122,179,180]

## Data Availability

No data were generated in this study.

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
