# Peer review of "Extracellular Matrix-Based and Electrospun Scaffolding Systems for Vaginal Reconstruction"

_bioengineering, 2023, doi:10.3390/bioengineering10070790_

Round 1
Reviewer 1 Report
Manuscript bioengineering-2399492
The manuscript is a non-systematic review that gives an overview of vaginal tissue engineering initiatives, mostly covering the authors´ experiences. The topics include: vaginal tissue morphology, vaginal physiology, extracellular matrix preparation in vitro, tissue de-cellularisation and electro-spinning.
The topics are presented in an educational way but the selection seems based on experiences by the own research group, and it is unclear why decellularized biological matrices and electrospinning was selected, but not other types of scaffold manufacturing (i.e. knitting, bioprinting, compression) and other specific types of ex vivo matrix production (besides the author’s self-reported self-assembly technique). Some limitations are mentioned but general issues, that are important for future patient implementation, are not covered. For instance, I miss reading about hardships that has lead difficulties related to clinical translation. As such, asection about regulatory safety measures would be appropriate and aspects related to time, costs and need of highly trained and dedicated laboratory personnel. These aspects would help the reader understand difficulties in relation to i.e. introducing the "self-assembly technique" to the clinic. Another aspect that could be interesting to discuss is patient selection; which are the patients that would be eligible when first introducing a new technique or material?
To give the manuscript a better flow, I would suggest either addressing vaginal organ reconstruction or vaginal prolapse, just because the preimplant needs in these two situations are quite different from a tissue engineering perspective. I also recommend omitting MRKHS as an example of an eligible patient group, as 1stline treatment with dilatations usually works, this patient group would not primarily benefit from tissue engineering techniques.
The production of the different methods is well described in the narrative, but the authors are encouraged to make some adjustments:
· Figure 1, it is an exact copy of another review article by the same authors. As far as I can see, only the figure legend differs. Please change to avoid self-plagiarism (or follow journal guidelines), and be more precise in what is copied in and what is not.
· Please also specify in this section, if the self-assembly method is success in recapitulating the complex biology and histology of the vaginal wall regarding extracellular matrix component composition, such as types of collagen fibers, elastin fibers and proteoglycans
· The last paragraph in 4.1 (page 5, lines 199-202) needs further clarification or a reference to the statement: in which way has the self-assembly method been combined with a scaffold?
· In 4.2 regarding tissue decellularization for generating ECM scaffold, it lacks examples of its use in vaginal tissue reconstruction. The table is far more extensive than the corresponding text, yet it seems to be quite generalizing regarding pros and cons. Some examples of studies looking at the tissue reaction on different decellularization processes could add substance to the text.
· 4.3 on electrospinning could also be strengthened, maybe by extending the figure legend and adding information on key points in the electrospinning technique, i.e. what is the function of the high positive voltage during production? thickness of fibers? are the feeders manually compressed? are they produced in sterile conditions or is that a next step?
· Could you address the issue of forming 3D scaffolds or sheets using electrospun biomaterials/ synthetic materials?
· Could extracellular matrix polymers such as collagens be subjected to electrospinning?
· Could you expand on the concept of hybrid scaffolds?
· It would be helpful to make a table with reference to scientific publications comparing the strength and weakness of each methodology regarding:
Time of production
Evidence of success and area of use (vaginal wall re-construction/replacement? re-enforcement?)
Model used (in vitro in vivo human or animal)
Similarity to native vagina
Weaknesses
Reported complications
· Section 5.1 presents some interesting details related to the self-assembly technique. The development of disease models is interesting, it could be introduced earlier and extended on.
· Page 13 (lines 473-474) need correction
Finally, I would appreciate a reflection on areas to improve to facilitate translation and a conclusion on which method that would be the most affordable and realistic for vaginal tissue reconstruction. Which are the characteristics that the authors think are desirable for an ideal vaginal scaffold? And what is “good enough” for introduction to the clinic?
Please see comments above.
Author Response
Reviewer 1:
The manuscript is a non-systematic review that gives an overview of vaginal tissue engineering initiatives, mostly covering the authors' experiences. The topics include vaginal tissue morphology, vaginal physiology, extracellular matrix preparation in vitro, tissue decellularization and electrospinning.
1- The topics are presented educationally, but the selection seems based on the experiences of the own research group. It is unclear why decellularized biological matrices and electrospinning was selected but not other types of scaffold manufacturing (i.e. knitting, bioprinting, compression) and other specific types of ex vivo matrix production (besides the author's self-reported self-assembly technique).
Response: We aimed to review the use of fibrous scaffolds that are either ECM-based or resemble ECM in structure or composition. That was the rationale behind selecting these scaffold fabrication methods for review. Lines 42-44 discuss this rationale.
“Significant strides have been made toward developing scaffolds that mimic vaginal ECM's fibrous structure or its composition. Different scaffold fabrication methods have been used, including electrospinning, self-assembly, and tissue acelularization”.
In addition, lines 48-50 also highlight the aim of this review.
“This review discusses the applications, challenges, and prospects of ECM-based scaffolds, including self-assembled tissues and decellularized scaffolds, as well as the electrospun constructs that mimic the architecture of vaginal ECM in the field of vaginal tissue engineering.”
2- Some limitations are mentioned, but general issues for future patient implementation are not covered. For instance, I miss reading about hardships that have led to difficulties related to clinical translation. As such, a section about regulatory safety measures would be appropriate, as aspects related to time, costs and the need for highly trained and dedicated laboratory personnel. These aspects would help the reader understand the difficulties of, i.e. introducing the "self-assembly technique" to the clinic. Another aspect that could be interesting to discuss is patient selection; which patients would be eligible when introducing a new technique or material?
Response: These limitations were further discussed in section 6.
3- To give the manuscript a better flow, I would suggest either addressing vaginal organ reconstruction or vaginal prolapse, just because the preimplant needs in these two situations are quite different from a tissue engineering perspective. I also recommend omitting MRKHS as an example of an eligible patient group, as 1stline treatment with dilatations usually works; this patient group would not primarily benefit from tissue engineering techniques.
Response: The rationale behind selecting both vaginal organ reconstruction and vaginal wall re-enforcement for this review was that tissue engineering strategies with the reviewed scaffold fabrication methods had been used for developing a potential cure in both cases. If we delete one of them, the number of studies would be minimal. However, MRKHS was deleted.
4- The production of the different methods is well described in the narrative, but the authors are encouraged to make some adjustments:
- Figure 1 is an exact copy of another review article by the same authors. As far as I can see, only the figure legend differs. Please change to avoid self-plagiarism (or follow journal guidelines), and be more precise in what is copied in and what is not.
Response: this figure was redesigned.
- Please also specify in this section if the self-assembly method is successful in recapitulating the complex biology and histology of the vaginal wall regarding extracellular matrix component composition, such as types of collagen fibres, elastin fibres and proteoglycans.
Response: We discussed this issue in the last paragraph of section 4-1.
The last paragraph in 4.1 (page 5, lines 199-202) needs further clarification or a reference to the statement: How has the self-assembly method been combined with a scaffold?
Response: This was merely a research idea we are working on in our group. We do not have a reference for this statement. Therefore, we deleted this sentence.
- In 4.2 regarding tissue decellularization for generating ECM scaffold, it lacks examples of its use in vaginal tissue reconstruction. The table is far more extensive than the corresponding text, yet it seems to be quite generalizing regarding the pros and cons. Some examples of studies looking at the tissue reaction on different decellularization processes could add substance to the text.
Response: In this section, we have provided a general overview of the current acelularization methods. Detailed discussions surrounding their application in tissue engineering have been well-reviewed elsewhere. These references were provided in the text. Some examples of tissue reaction to different decellularization methods were summarized in the text (last paragraph of section 4-2).
- 4.3 on electrospinning could also be strengthened, maybe by extending the figure legend and adding information on key points in the electrospinning technique, i.e. what is the function of the high positive voltage during production? The thickness of fibres? Are the feeders manually compressed? Are they produced in sterile conditions, or is that the next step?
Response: More information was provided in the figure legend.
- Could you address the issue of forming 3D scaffolds or sheets using electrospun biomaterials/ synthetic materials?
Response: More information was provided at the end of section 4-3.
- Could extracellular matrix polymers such as collagens be subjected to electrospinning?
Response: The ECM-derived polymers need a driving polymer to be electrospun. They are typically blended with synthetic polymers for electrospinning, such as PCL, PLLA, PU, etc.
- Could you expand on the concept of hybrid scaffolds?
Response: More information was provided at the end of section 4-3.
- It would be helpful to make a table with reference to scientific publications comparing the strength and weaknesses of each methodology regarding:
Time of production
Evidence of success and area of use (vaginal wall reconstruction/replacement? re-enforcement?)
The model used (in vitro, in vivo human or animal)
Similarity to the native vagina
Weaknesses
Reported complications
Response: A new table (Table 3) was incorporated. However, limited data are available regarding the complications. This item was replaced with strengths.
- Section 5.1 presents some interesting details related to the self-assembly technique. The development of disease models is interesting; it could be introduced earlier and extended.
Response: This method was further explained in paragraph 3 of section 4-1.
Page 13 (lines 473-474) needs correction.
Response: "There new strategies for vaginal luminal surface epithelization should be developed to prevent lumen loss" was replaced with "Therefore, new strategies should be developed to promote epithelization of the vaginal luminal surface and prevent the loss of the lumen."
- Finally, I would appreciate a reflection on areas to improve to facilitate translation and a conclusion on which method would be the most affordable and realistic for vaginal tissue reconstruction. Which characteristics do the authors think are desirable for an ideal vaginal scaffold? And what is “good enough” for introduction to the clinic?
Response: Our opinion was added in the last paragraph of section 6.
Reviewer 2 Report
Review of manuscript ID: bioengineering-2399492 entitled ‘’ Extracellular matrix-based and electrospun scaffolding systems for vaginal reconstruction’’.
The above-titled manuscript is a review paper that discusses recent applications, challenges, and future perspectives of constructs in vaginal reconstruction strategies. The authors argue that current surgical and non-surgical treatments for vaginal defects are available but are limited in their efficacy and they often result in long-term complications. In this review, the authors present evidence that tissue-engineered neovagina is a promising new approach that provides an extracellular matrix (ECM)-like environment for vaginal cells to adhere, secrete ECM, and remodel by host cells. To this end, ECM-based and electrospun scaffolding systems, generated by self-assembly, electrospinning, or decellularization techniques, have gained attention from both clinicians and researchers. These biomimetic scaffolds are highly similar to the native vaginal ECM and have great potential for clinical translation.
Overall, the manuscript is written well but requires revisions before consideration for publication.
Specific Points
1. Introduction- the introduction must start by introducing the general subject of regenerative medicine and tissue engineering, which have shown the ability to influence and impact patients’ treatment via strategies that can restore or improve tissues and organ functions. A whole paragraph must be devoted to this before the authors introduce the issue of Congenital Vaginal Anomalies (CVA). CVA is just an example of conditions inflicting humanity and can be addressed via regenerative medicine and tissue engineering. Various well-written manuscripts are available on this topic and must include:
PMID: 34327664
PMID: 29990578
PMID: 31540457
PMID: 30154861
PMID: 12435034
PMID: 27398431
PMID: 29990578
PMID: 32463143
2. Table 1 is good except that it’s a general discussion of methods used in the decellularization of tissues. Authors must give examples of decellularization methods used for vaginal tissues. The information in Table 1 is not new and can only be novel if it presents work done on vaginal tissues.
3. The section on Brief histology of the vagina and the biology of its ECM is OK. But this must be followed by a Discussion of how attempts at mimicking the ECM of the vagina are limited or face challenges etc. This section must be expanded.
4. The section on Challenges, future perspectives, and concluding remarks must be expanded to a minimum of 2 pages. Future perspectives must present new ideas being explored on this topic and the advantages and disadvantages.
5. This is a review article and must acknowledge work done or written by other scientists already. Therefore, References must be a minimum of 200 acceptable for this review.
6. There are many instances of spelling and grammar mistakes. Revise the whole manuscript.
7. Authors present only 2 Figures and 1 Table. For a review on regenerative medicine and bioinks (decellularization and polymers), this is unacceptable. More Figures and Tables are required to describe many of the concepts presented in this manuscript.
Review of manuscript ID: bioengineering-2399492 entitled ‘’ Extracellular matrix-based and electrospun scaffolding systems for vaginal reconstruction’’.
The above-titled manuscript is a review paper that discusses recent applications, challenges, and future perspectives of constructs in vaginal reconstruction strategies. The authors argue that current surgical and non-surgical treatments for vaginal defects are available but are limited in their efficacy and they often result in long-term complications. In this review, the authors present evidence that tissue-engineered neovagina is a promising new approach that provides an extracellular matrix (ECM)-like environment for vaginal cells to adhere, secrete ECM, and remodel by host cells. To this end, ECM-based and electrospun scaffolding systems, generated by self-assembly, electrospinning, or decellularization techniques, have gained attention from both clinicians and researchers. These biomimetic scaffolds are highly similar to the native vaginal ECM and have great potential for clinical translation.
Overall, the manuscript is written well but requires revisions before consideration for publication.
Specific Points
1. Introduction- the introduction must start by introducing the general subject of regenerative medicine and tissue engineering, which have shown the ability to influence and impact patients’ treatment via strategies that can restore or improve tissues and organ functions. A whole paragraph must be devoted to this before the authors introduce the issue of Congenital Vaginal Anomalies (CVA). CVA is just an example of conditions inflicting humanity and can be addressed via regenerative medicine and tissue engineering. Various well-written manuscripts are available on this topic and must include:
PMID: 34327664
PMID: 29990578
PMID: 31540457
PMID: 30154861
PMID: 12435034
PMID: 27398431
PMID: 29990578
PMID: 32463143
2. Table 1 is good except that it’s a general discussion of methods used in the decellularization of tissues. Authors must give examples of decellularization methods used for vaginal tissues. The information in Table 1 is not new and can only be novel if it presents work done on vaginal tissues.
3. The section on Brief histology of the vagina and the biology of its ECM is OK. But this must be followed by a Discussion of how attempts at mimicking the ECM of the vagina are limited or face challenges etc. This section must be expanded.
4. The section on Challenges, future perspectives, and concluding remarks must be expanded to a minimum of 2 pages. Future perspectives must present new ideas being explored on this topic and the advantages and disadvantages.
5. This is a review article and must acknowledge work done or written by other scientists already. Therefore, References must be a minimum of 200 acceptable for this review.
6. There are many instances of spelling and grammar mistakes. Revise the whole manuscript.
7. Authors present only 2 Figures and 1 Table. For a review on regenerative medicine and bioinks (decellularization and polymers), this is unacceptable. More Figures and Tables are required to describe many of the concepts presented in this manuscript.
Author Response
Reviewer 2:
The above-titled manuscript is a review paper that discusses recent applications, challenges, and future perspectives of constructs in vaginal reconstruction strategies. The authors argue that current surgical and non-surgical treatments for vaginal defects are available but are limited in efficacy, often resulting in long-term complications. In this review, the authors present evidence that tissue-engineered neovagina is a promising new approach that provides an extracellular matrix (ECM)-like environment for vaginal cells to adhere, secrete ECM, and remodel by host cells. To this end, ECM-based and electrospun scaffolding systems generated by self-assembly, electrospinning, or decellularization techniques, have gained attention from both clinicians and researchers. These biomimetic scaffolds are highly similar to the native vaginal ECM and have great potential for clinical translation.
Overall, the manuscript is written well but requires revisions before consideration for publication.
Specific Points
- Introduction- the introduction must start by introducing the general subject of regenerative medicine and tissue engineering, which have shown the ability to influence and impact patients' treatment via strategies that can restore or improve tissues and organ functions. A whole paragraph must be devoted to this before the authors introduce the issue of Congenital Vaginal Anomalies (CVA). CVA is an example of conditions inflicting humanity and can be addressed via regenerative medicine and tissue engineering. Various well-written manuscripts are available on this topic and must include the following:
PMID: 34327664
PMID: 31540457
PMID: 30154861
PMID: 12435034
PMID: 29990578
PMID: 32463143
Response: The introduction section was revised, and most papers were cited.
- Table 1 is good, except it's a general discussion of methods used to decellularize tissues. Authors must give examples of decellularization methods used for vaginal tissues. The information in Table 1 is not new and can only be novel if it presents work done on vaginal tissues.
Response: Examples of decellularized scaffolds have been summarized in section 5-3. If the reviewers think this table is redundant, it could be deleted.
- The section on the Brief histology of the vagina and the biology of its ECM is OK. But this must be followed by a Discussion of how attempts at mimicking the ECM of the vagina are limited or face challenges etc. This section must be expanded.
Response: This section was expanded in the last paragraph of section 3.
- The section on Challenges, future perspectives, and concluding remarks must be expanded to a minimum of 2 pages. Future perspectives must present new ideas being explored on this topic and the advantages and disadvantages.
Response: This section was expanded, and new ideas were incorporated.
- This is a review article and must acknowledge work done or written by other scientists already. Therefore, References must be a minimum of 200 acceptable for this review.
Response: We cited more papers. We initially limited the number of references to avoid redundancy. The number of citations was increased to 208.
- There are many instances of spelling and grammar mistakes. Revise the whole manuscript.
Response: The whole manuscript was checked for spelling and grammatical mistakes.
- Authors present only 2 Figures and 1 Table. For a review on regenerative medicine and bioinks (decellularization and polymers), this is unacceptable. More Figures and Tables are required to describe many of the concepts presented in this manuscript.
Response: We added two more tables and figures. However, we cannot add images from the original research papers due to budget limitations.
Reviewer 3 Report
In this article, the authors reviewed the various methods of regenerating vaginal ECM. They reviewed the pertinent articles in literature focusing primarily on electrospun scaffolds. The article was written well overall, although a few typos and incomplete sentences should be edited. Here are some minor comments:
Line 37: I wouldn't call it a "disease". Perhaps a disorder or medical condition?
Lines 76-77: incomplete sentence.
Line 88: replace "for" with "by".
Line 127: "latter".
Line 193: "adapted".
Line 257: what is "(s2)"?
Lines 397-398: any reason for large font?
Line 473: "There"?
Line 481: "theses".
Lines 492-493: sentence incomplete.
The real contribution of this submission beings at Section 5. It would have been ideal if the authors included some relevant images (histological, SEM, etc.) from literature and some quantitative data on mechanical properties, physico-chemical characteristics, etc.
My comments above highlight some of the typos and incomplete sentences. Perhaps the authors could do a thorough proof-reading before resubmission.
Author Response
Reviewer 3:
In this article, the authors reviewed the various methods of regenerating vaginal ECM. They reviewed the pertinent articles in the literature focusing primarily on electrospun scaffolds. The article was written well overall, although a few typos and incomplete sentences should be edited. Here are some minor comments:
Line 37: I wouldn't call it a "disease." Perhaps a disorder or medical condition?
Response: The change has been made; we have replaced disease with a medical condition.
*Lines 76-77: incomplete sentence.
Responses: this sentence was rewritten.
Line 88: replace "for" with "by."
Responses: "For" was replaced with "By."
Line 127: "latter."
Responses: We corrected this word.
Line 193: "adapted."
Response: Done
Line 257: what is "(s2)"?
Response: S2 was removed.
Lines 397-398: any reason for the large font?
Response: The font was adjusted.
Line 473: "There"?
Response: There was replaced by, therefore.
Line 481: "theses."
Response: Theses replaced by these
Lines 492-493: sentence incomplete.
Response: this sentence was rewritten.
The real contribution of this submission beings in Section 5. It would have been ideal if the authors included some relevant images (histological, SEM, etc.) from the literature and some quantitative data on mechanical properties, physicochemical characteristics, etc.
Response: Some figures were added from the reviewed papers. However, we could not afford to add other figures to the manuscript due to budget limitations (paying for the use of copyright figures).
Comments on the Quality of English Language
My comments above highlight some of the typos and incomplete sentences. Perhaps the authors could do a thorough proofreading before resubmission.
Response: A thorough proofreading was performed to improve the English content.
Round 2
Reviewer 2 Report
The manuscript has been improved to suit publication standard
The manuscript is greatly improved.